# Stable Nitroxide as Diagnostic Tools for Monitoring of Oxidative Stress and Hypoalbuminemia in the Context of COVID-19

**DOI:** 10.3390/ijms25158045

**Published:** 2024-07-24

**Authors:** Ekaterina Georgieva, Julian Ananiev, Yovcho Yovchev, Georgi Arabadzhiev, Hristo Abrashev, Vyara Zaharieva, Vasil Atanasov, Rositsa Kostandieva, Mitko Mitev, Kamelia Petkova-Parlapanska, Yanka Karamalakova, Vanya Tsoneva, Galina Nikolova

**Affiliations:** 1Department of General and Clinical Pathology, Forensic Medicine, Deontology and Dermatovenerology, Medical Faculty, Trakia University, 11 Armeiska Str., 6000 Stara Zagora, Bulgaria; ekaterina.georgieva@trakia-uni.bg (E.G.); julian.ananiev@trakia-uni.bg (J.A.); vyara.zaharieva@trakia-uni.bg (V.Z.); 2Department of Surgery and Anesthesiology, University Hospital “Prof. Dr. St. Kirkovich”, 6000 Stara Zagora, Bulgaria; yovcho.yovchev@trakia-uni.bg (Y.Y.); georgi.arabadzhiev@trakia-uni.bg (G.A.); 3Department of Vascular Surgery, Medical Faculty, Trakia University, 11 Armeiska Str., 6000 Stara Zagora, Bulgaria; hristo.abrashev@trakia-uni.bg; 4Forensic Toxicology Laboratory, Military Medical Academy, 3 G. Sofiiski, 1606 Sofia, Bulgaria; vatanasov@chem.uni-sofia.bg (V.A.); toxilab@vma.bg (R.K.); 5Department of Diagnostic Imaging, University Hospital “Prof. Dr. St. Kirkovich”, 6000 Stara Zagora, Bulgaria; mitko.mitev@trakia-uni.bg; 6Department of Medical Chemistry and Biochemistry, Medical Faculty, Trakia University, 11 Armeiska Str., 6000 Stara Zagora, Bulgaria; kamelia.parlapanska@trakia-uni.bg (K.P.-P.); galina.nikolova@trakia-uni.bg (G.N.); 7Department of Propaedeutics of Internal Medicine and Clinical Laboratory, Medical Faculty, Trakia University, 11 Armeiska Str., 6000 Stara Zagora, Bulgaria; v.tsoneva.ivanova@trakia-uni.bg

**Keywords:** COVID-19, albumin, hypoalbuminemia, oxidative stress, ROS, nitroxide radicals, EPR spectroscopy, SARS-CoV-2 infections

## Abstract

Oxidative stress is a major source of ROS-mediated damage to macromolecules, tissues, and the whole body. It is an important marker in the severe picture of pathological conditions. The discovery of free radicals in biological systems gives a “start” to studying various pathological processes related to the development and progression of many diseases. From this moment on, the enrichment of knowledge about the participation of free radicals and free-radical processes in the pathogenesis of cardiovascular, neurodegenerative, and endocrine diseases, inflammatory conditions, and infections, including COVID-19, is increasing exponentially. Excessive inflammatory responses and abnormal reactive oxygen species (ROS) levels may disrupt mitochondrial dynamics, increasing the risk of cell damage. In addition, low serum albumin levels and changes in the normal physiological balance between reduced and oxidized albumin can be a serious prerequisite for impaired antioxidant capacity of the body, worsening the condition in patients. This review presents the interrelationship between oxidative stress, inflammation, and low albumin levels, which are hallmarks of COVID-19.

## 1. Introduction

Clinical and pathoanatomical studies point to the cytokine storm as a significant factor in the development of acute respiratory distress syndrome (ARDS) and subsequent multiple organ failure in patients with COVID-19 [1]. A systemic inflammatory response known as cytokine release syndrome or cytokine storm plays a major role in developing acute injury during COVID-19. A rapid increase in the levels of pro-inflammatory cytokines in SARS-CoV-2 infection is associated with the development of acute respiratory distress syndrome (ARDS) [2]. In severe COVID-19, ARDS is defined as a predictable serious complication that requires early recognition. The entry of the SARS-CoV-2 virus into the respiratory system and the subsequent strong inflammatory response leads to the destruction of the alveolar-capillary barrier. In the acute stage of infection, ARDS causes lung damage, which includes the formation of hyaline membranes in the alveoli, followed by interstitial expansion, fibroblast proliferation, and typical pathological changes characterized by diffuse alveolar parenchymal damage and edema [3]. Diffuse alveolar damage results in impaired gas exchange, with refractory hypoxemia and hypercarbia, intrapulmonary shunt, and reduced functional lung surface [4]. Accumulation of fluid in the alveolar and interstitial spaces causes inhibition of pulmonary surfactant [5]. The lung morphology is characterized by a rapid evolution from interstitial and alveolar edema to fibrosis as a result of damage to the alveolar-capillary unit [6]. Viral infections are known to be characterized by the production of abnormally high levels of oxygen radicals. For example, SARS-CoV-2 causes over activation of the immune response in lung tissues, which is almost always accompanied by oxidative stress and subsequent endothelial damage [7]. In addition to severe ARDS and uncontrolled inflammation, SARS-CoV-2 infection is associated with excessive neutrophil extracellular trap formation and subsequent vascular damage [8]. SARS-CoV-2 activates platelets, leading to endothelial dysfunction, and apparent stasis and hypercoagulation predispose patients with severe organ dysfunction to thrombotic events involving arterial and venous vascular areas [9,10].

The high degree of lung damage during SARS-CoV-2 infection generates large amounts of reactive oxygen and nitrogen species (ROS, RNS) and the release of pro-inflammatory cytokines from monocytes and macrophages. SARS-CoV-2 can compromise mitochondrial dynamics through mitochondrial DNA damage, changes in mitochondrial membrane potential, and calcium homeostasis, disrupting the redox balance in the organism during infection progression [11,12]. On the one hand, COVID-19 lung tissue damage leads to critical levels of free radicals, and on the other hand, hyperoxia induces the overproduction of mitochondrial ROS. An imbalance of the intracellular redox state and activation of redox-sensitive effector pathways is observed, followed by a strong immune response, apoptosis, and necrosis as a body systemic inflammatory response [13]. Excess ROS causes irreversible oxidative damage to important biomacromolecules, membrane phospholipids, and proteins [14]. The severe inflammatory state in COVID-19 is accompanied by an abnormal immune response, marked lymphopenia, and high levels of D-dimer, cytokines, C-reactive protein, etc. [15]. The clinical manifestation is complemented by another important prognostic marker—serum albumin. High levels of pro-inflammatory cytokines initiate the release of large amounts of hepcidin, leading to the disruption of iron homeostasis [16]. Paliogiannis and colleagues indicated that the decrease in human serum albumin (HSA) concentrations may be considered one of the most striking signs of an advanced SARS-CoV-2 infection [17]. HSA is affected by various factors, such as changes in the extracellular redox balance in favor of oxidants, which can lead to structural disorders in its molecule, pH, and transport of transition metal ions, nitric oxide, hemin, and drugs. The overproduction of free radicals, together with an increased immune response and reduced endogenous enzymatic and non-enzymatic antioxidant defenses, lead to a vicious cycle of mutually induced hyperinflammation and massive oxidative damage during the acute phase of COVID-19. This implies a high degree of structural changes in macromolecules, the formation of non-functional protein derivatives, and a high degree of damage to cellular components [7].

The high levels of ROS and RNS generated during the acute phase of COVID-19 can lead to irreversible oxidative damage to albumin and impairment of its antioxidant properties [18]. This suggests that redox imbalance, together with reduced albumin levels, increases the risk of mortality in patients with SARS-CoV-2 infection [19,20,21,22]. Therefore, albumin values below 3.5 g/dL or 35 g/L might be considered as an indicator determining the severity of SARS-CoV-2 infection and disease outcome [23] (Figure 1).

The reported abnormal levels of ROS and OS suggest a high degree of structural changes in the protein molecule, formation of non-functional protein derivatives, and high concentration of oxidized albumin in COVID-19. The high levels of ROS, the oxidized form of albumin, and low levels of protein can make it impossible to obstruct COVID-19 induced oxidative injury and lead to large-scale organ damage [24].

## 2. Methods

### 2.1. Research Strategy

We used the databases PubMed, Scopus, MDPI platform, WoS, Google Scholar, and ResearchGate for the literature search, among which over 2390 scientific papers were identified and reviewed. The methodology, data extraction, and synthesis were adapted to the PRISMA-P [25] and PRISMA-S guidelines criteria [26]. This review employs a narrow-specialized scope of science to describe COVID-19-related hypoalbuminemia. First, it is aimed at hypoalbuminemia in SARS-CoV-2 infection, and second, at nitroxide radicals and EPR spectroscopy as a future diagnostic tool based on the assessment of total oxidative stress, ROS-induced qualitative changes in albumin, and quantitative changes in serum protein levels in COVID-19. The research strategy included data review and analysis of existing research and review articles in the timeframe from 2002 to 2024 (Figure 2).

### 2.2. Inclusion and Exclusion Criteria

#### 2.2.1. Inclusion Criteria

The scientific articles, systematic reviews, and meta-analyses in the last years were used and included predominantly from 2002 to those published on 15 June 2024 since the study aimed to emphasize the most recent evidence. In comprehensive electronic research, we use the following keywords and different combinations of those (all fields): oxidative stress, free radicals, ROS and RNS, reactive oxygen and nitrogen species (RONS), hydroxyl radical, hydrogen peroxide, Fenton reaction, redox state, superoxide radical, radical generation, inflammation, cytokines, SARS-CoV-2 infection, COVID-19, albumin, transport protein, low albumin levels, hypoalbuminemia, diseases, EPR spectroscopy, nitroxide radicals, SDSL-EPR spectroscopy, TEMPOL, 3-Maleimido-Proxyl radical (5-MSL).

#### 2.2.2. Exclusion Criteria

Our research did not include case reports, preprint studies, comments, letters to editors, and scientific research involving in vitro experiments (cells) and animals. We excluded all research that was written in a language other than English.

## 3. Results and Discussion

### 3.1. The Oxidative Species Generation in Healthy Organisms and Antioxidant Defense System

At physiological levels, ROS function as redox mediators, participating in signal transduction and promoting cell proliferation and cell survival, while high levels of ROS can induce cell death [27]. Depending on the concentration of ROS, it can be beneficial or harmful to cells and tissues. A healthy organism is in a state of equilibrium, characterized by the maintenance of physiological levels of ROS by intracellular reductors. Under physiological conditions, the balance between ROS production and scavenging is the main factor responsible for maintaining redox homeostasis, ensuring that cells will respond properly to endogenous and exogenous stimuli. With the so-called “steady state”, intracellular ROS levels are tightly regulated by antioxidant enzymes that maintain cellular redox homeostasis [28]. However, disturbances in redox homeostasis and induction of oxidative stress lead to some pathological processes, abnormal cell death, and the development of various diseases. Oxidative stress serves not only as a type of stimulus to induce a transduction response but can also modulate apoptosis through direct modifications of biological macromolecules. When oxidative stress occurs, cells try to counteract it by restoring the redox balance and regulating critical homeostatic parameters. Such cellular activity results in the activation or inactivation of genes encoding defense enzymes, transcription factors, and structural proteins [29]. ROS are involved in cell signaling in the regulation of cellular processes, thanks to their ability to mediate the reversible oxidation of cysteine. For example, H_2_O_2_ has emerged as the major redox-signaling metabolite capable of mediating the reversible oxidation of thiol groups in proteins [27].

#### 3.1.1. Superoxide Anion Radical Generation

The superoxide anion radical (O_2_•^−^) is generated in cells by enzymatic and non-enzymatic processes. Enzymatic sources include NADPH oxidase activity, xanthine oxidase, the mitochondrial electron transport chain, and some enzymes in peroxisomes [30]. Non-enzymatic sources can be redox reactions involving metals such as iron and copper and external sources, such as toxins and xenobiotics radiation, and some drugs. The superoxide anion radical can react with another superoxide to generate hydrogen peroxide (H_2_O_2_), which can be reduced to water or partially reduced to the highly reactive hydroxide radical (•OH) [31]. In turn, superoxide anion dismutation can be spontaneous or catalyzed by enzymes known as superoxide dismutases (SODs). The formation of •OH is possible by the decomposition of H_2_O_2_ in the presence of ions of transition metals in a lower valence state (Fe^2+^ or Cu^+^). An important mechanism for the generation of hydroxide radicals is the reaction of hydrogen peroxide and superoxide radicals (Haber–Weiss reaction) in the presence of oxidized transition metals [32,33,34].

#### 3.1.2. Role of Mitochondria

Mitochondrial dysfunction has been proposed as a potential mechanism in the pathology of COVID-19. Mitochondrial redox control is important not only for oxidative phosphorylation, ATP synthesis, calcium homeostasis, thermogenesis, apoptosis, and ROS production but also for maintaining redox balance in cells. Mitochondrial gene mutations, which underlie various diseases, can disrupt mitochondrial energy metabolism, mitochondrial bioenergetics, and biosynthesis and serve as a trigger for mitochondrial “retrograde signaling” in the nucleus. Disorders in redox control define mitochondria as the main source of intracellular oxidants [7,35]. Pathological changes in mitochondrial dynamics can be caused by the overproduction of ROS and the mitochondrial dysfunction initiated by them. As a result, processes of oxidative DNA damage are promoted, and a prerequisite is created for impaired redox regulation, which contributes to a wide range of pathological changes in cells [36]. Several studies have reported that SARS-CoV-2 can lead to mitochondrial dysfunction in various cell types. The virus induces a significant decrease in mitochondrial membrane potential and increased ROS production in lung epithelial cells, which may lead to OS and lung tissue damage [35]. Direct evidence for this is found to be higher levels of mtDNA in the blood of patients with COVID-19, which is due to increased mitochondrial stress (mtOS) and mitochondrial dysfunction [7]. The redox couple GSH/GSSG is fundamental for cells and plays a key role in the regulation of redox-dependent cellular functions—through thiol modifications. As a key modulator of cellular functions, GSH is involved in cellular defense against oxidative damage, in the redox regulation of protein thiols and maintenance of cellular redox homeostasis in nutrient metabolism, and in the regulation of cellular metabolic functions ranging from gene expression, protein synthesis, signal transduction to cell proliferation, and apoptosis [37]. Any changes in the redox balance of the cell in favor of the oxidized form of glutathione–GSSG represents an important signal that can determine the fate of the cell [38]. Depletion of mitochondrial GSH can lead to increased release of H_2_O_2_ from the matrix, which can oxidize cytoplasmic proteins and affect cell signaling [39]. In conditions of prolonged oxidative stress, when cellular defense systems cannot counteract oxidative-mediated damage, the amount of free GSH decreases. This leads to irreversible cell degeneration and death [40].

Reduced plasma GSH levels have been observed in patients with COVID-19. The increased oxidation of cellular components such as the ROS/GSH ratio in favor of oxidants strongly correlates with the severity, symptoms, and recovery period of infection [41]. Protein glutathionylation is the main redox immune mechanism that prevents or alleviates damage to cellular components [42]. In patients with COVID-19, systemic oxidative stress leads to a decrease in GSH levels, promoting the development of infection, while an increase in GSH levels inhibits disease progression. Despite the increased activity of enzymes, especially glutathione peroxidase, low levels of GSH inhibit the effectiveness of non-enzymatic antioxidant activity, especially in those who died as a result of COVID-19 [41,42,43]. NADPH is an important cofactor involved in many physiological processes and disturbances in its synthesis, as well as an imbalance in the ratio of reduced/oxidized forms leads to the development of pathologies. In the mitochondria, the redox couple NADPH/NADP+ under the action of nicotinamide nucleotide transhydrogenase in combination with NADH/NAD+ leads to the accumulation of NADPH. Any disturbance in the components of this enzyme could lead to changes in the redox state and the appearance of oxidative damage in mitochondrial structures. NADPH-dependent oxidases use NADPH as a cofactor and are thought to be the only group of enzymes for which ROS production is a major function. They are membrane-bound enzymes that generate superoxide and other ROS (H_2_O_2_) at the plasma membrane. Disturbances in NADPH production are expected to modulate cellular redox balance and lead to increased oxidation [44]. Increased oxidative stress through Nox2 activation is associated with severe disease and thrombotic events in COVID-19. ROS production and S protein-induced activation of NOX2 in endothelial cells initiate endothelial dysfunction in cardiac microvascular endothelium in deceased patients with COVID-19 [45]. SARS-CoV-2 generates downregulation of angiotensin-converting enzyme 2 (ACE2) and transmembrane protease serine 2 (TMPRSS2) receptors, reducing their number, leading to increased activation of angiotensin II and decreased levels of angiotensin. Binding of the virus to the ACE2 receptor causes excessive release of various inflammatory cytokines, followed by disturbances in the regulation of the renin–angiotensin–aldosterone system, activation of NADPH oxidase, progression of infection, and coagulation disorders [46].

#### 3.1.3. The Role of Endogenous Antioxidants

The body’s defense against oxidative damage involves strict redox control and maintenance of a delicate balance between the formation and elimination of ROS and RNS [47]. To counteract the harmful effects of free radicals, the body has an endogenous antioxidant defense system, including SOD, CAT, GPx, and glutathione reductase [29]. SOD catalyzes the conversion of the highly reactive superoxide radical (O_2_•^−^) to the less reactive hydrogen peroxide through redox reactions involving metal ions of variable valence [48]. Catalase is a hemoprotein involved in the metabolism of hydrogen peroxide, degrading it to water and molecular oxygen. Catalase shares this function with glutathione peroxidases. Other enzymes involved in ROS scavenging are glutathione-S-transferases (GSTs) [49]. They are involved in the metabolism of xenobiotics and the synthesis of some endogenous biologically important compounds. Antioxidants such as GSH and the endogenous antioxidants SOD, CAT, and GPX play a leading role in preventing oxidative damage in COVID-19. In patients with SARS-CoV-2 and high levels of ROS, a higher activity of CAT and SOD enzymes was observed, which correlated with the severity of the disease. At the same time, low levels of reduced thiol and reduced total antioxidant capacity are observed, especially in hospitalized patients and those with a severe form of infection [50].

### 3.2. Oxidative Stress and COVID-19

In COVID-19, overproduction of oxidants, heightened immune response, and reduced endogenous enzymatic and non-enzymatic antioxidant defenses result in a vicious cycle of mutually induced hyperinflammation and massive oxidative damage. Excessive inflammatory responses and abnormal ROS levels during the acute phase of infection can disrupt mitochondrial dynamics, increasing the risk of fatal COVID-19 [51]. Selective expression of specific genes is a central player in the adaptive response to OS and the reversible oxidation of cysteine residues in many proteins [52]. Iron (Fe) is involved in a wide variety of metabolic processes, including oxygen and electron transport and DNA synthesis. Iron ions Fe^2+^ and Fe^3+^ undergo reversible oxidation and reduction reactions, making them essential factors in electron transfer processes in biological systems [51]. The redox-active form of Fe^3+^ participates in a redox cycling reaction, leading to the generation of ROS. This involves electron transfer between Fe ions and other molecules, leading to the production of reactive intermediates and ROS (Fenton and Haber–Weiss) [32]. Transition metal-mediated ROS generation causes DNA modifications, increased lipid peroxidation, and altered calcium and sulfhydryl homeostasis. In turn, the lipid peroxides formed can further react with redox metals to form the mutagenic and carcinogenic malondialdehyde, 4-hydroxynonenal, and other exocyclic adducts [33]. Hydrogen peroxide and neutrophil-mediated OS are thought to be factors inducing structural changes in HSA in critically ill patients with COVID-19 [34]. Cavalcanti et al. [53] looked at H_2_O_2_ and its conversion to the highly reactive •OH in severe and critical COVID-19. Indeed, OC and high levels of H_2_O_2_ lead to an increased concentration of oxidized albumin, which is the less flexible and more rigid form of the protein, and a loss of biological activity of the protein.

Nitric oxide (NO) plays a major role as a signaling molecule in the regulation of vascular tone and blood flow. It is also involved in tissue oxygenation by controlling mitochondrial O_2_ consumption by inhibiting cytochrome c oxidase [54]. In addition to endothelial cells, NO can be generated by immune cells in response to inflammatory stimuli, leading to protection against infections. This also determines its function as a regulatory molecule in the modulation of inflammatory reactions and the activation of the body’s immune response to pathogens [55]. The reactivity of NO is quite limited and, therefore, its direct toxicity to the organism is less than that of most ROS. However, nitric oxide reacts with O_2_•^−^ to the highly aggressive and toxic proteins, lipids, and DNA, while the peroxynitrite anion (ONOO^−^) causes disruption of redox signaling and compromises the maintenance of cellular redox balance. Redox-active species derived from NO react rapidly with fatty acids and lipids, generating oxidized and nitrated products of free lipids and esterified cholesterol [28]. Peroxynitrite (ONOO^−^) is a highly reactive powerful oxidizing and nitrating agent. In biological systems, it is formed by a rapid diffusion-controlled reaction between nitric oxide and O_2_•^−^ [56]. The presence of transition metals, such as iron or copper, can speed up the process. It is characterized by strong reactivity, which through oxidation, nitration, and modification of various biomolecules causes nitrosative stress in various types of cells. Nitration of proteins can initiate structural modifications and alter the activity of numerous biomolecules, contributing to cellular dysfunction and disruption of normal physiological processes in the body [57].

In severe COVID-19, IL-1 and IL-2 are the main interleukins involved in RNS production. The main reactive nitrogen species (RNS) that have been identified as mediators of redox reactions in patients with COVID-19 are nitric oxide (NO•), nitrogen dioxide radical (ONO•), and peroxynitrite radical (ONOO•). During the cytokine storm, the induced enzyme nitric oxide synthetase (NOs) stimulates the generation of nitric oxide. The NO formed reacts with the superoxide anion, resulting in the formation of the highly reactive peroxynitrite radical [58]. Peroxynitrite has a major role in tissue damage and inflammation in various diseases. It participates in inflammation by modulating the function of immune cells and the release of pro-inflammatory cytokines and chemokines [36]. The mechanism involves the parallel formation of large amounts of “inflammatory NO” and O_2_•^−^, which together form ONOO^−^, which initiates DNA damage, lipid oxidation, tyrosine nitration, and inhibition of mitochondrial respiration [59].

These findings highlight the development of new methods for monitoring the COVID-19-dependent hypoalbuminemia and protein dysfunction, which Aziz and colleagues suggest is a hallmark of worsening general health and increased risk of mortality in patients with SARS-CoV-2 infection [60]. In excessive amounts, iron ions can cause cell and tissue damage, so their concentration in body tissues must be tightly regulated [61]. Non-transferrin-bound iron (NTBI) is the most harmful form of free iron in the blood, which is involved in ROS and RNS generation and oxidative stress in humans [62]. In addition, the overproduction of free radicals initiates conformational changes in proteins such as albumin, impairing their function [63]. The direct involvement of the protein in scavenging ROS and limiting oxidative damage in the body can be compromised by low concentrations of reduced albumin (redHSA) [64].

### 3.3. Albumin

#### 3.3.1. Physiological Functions of Albumin

Albumin (HSA) is the most abundant plasma protein in the human body with a molecular weight of approximately 66 kDa and a half-life in the body of 21 days. HSA is a globular, core-shaped protein that consists of three structurally homologous helical domains (I, II, and III), each comprising two subdomains, A and B [65]. Every single chain includes 585 amino acids as the first three aspartates, alanine, and histidine (Asp-Ala-His), which serves as a specific binding site for transition metals such as iron(II), cobalt (II), copper (II), and nickel (II) [66]. Molecular dynamics analysis of HSA shows that the movements of domains I and III are keys in determining the protein properties [67,68]. In healthy humans, HSA concentration ranges from 35 to 50 g/L and accounts for 50% of total plasma protein content [69]. Synthesis takes place in the polysomes of hepatocytes, which produce 10–15 g of protein per day. A small amount of albumin is stored in the liver (about 10%), and most of it is rapidly released into the blood [70]. Synthesis of HSA is stimulated by hormones such as insulin and growth hormone [71].

#### 3.3.2. Albumin as Colloid Osmotic Pressure Regulatory Molecule

Albumin has several physiological roles, one of the most important being the maintenance of colloid osmotic pressure in the vascular compartments, preventing fluid leakage into the extravascular spaces [72]. After albumin enters the circulation, about 30% remains in the bloodstream, and the remainder enters the interstitial space. The osmotic effect of albumin is due to its large molecular weight and its negative charge, which allows it to attract positively charged molecules and water into intravascular compartments. Low albumin concentrations increase vascular permeability to cells and plasma solutes and complicate critical and chronic conditions [70,73].

#### 3.3.3. Transport Function of Albumin

Albumin is the most abundant transport protein in blood plasma, which is responsible for the reversible binding of numerous endogenous and exogenous compounds. Its high overall binding capacity defines it as a major transporter of nutrients and neutral lipophilic and acidic dosage forms [74]. HSA is the main carrier of bilirubin, fatty acids drugs, circulating calcium, and various hormones (thyroxine, cortisol, testosterone, etc.). Ligand binding to the protein occurs mainly in subdomains IIA and IIIA. When albumin interacts with various substances, cooperativity, and allosteric modulation effects occur, which are usually inherent to multimeric macromolecules [67]. Of particular importance is not only the determination of the albumin concentration but also the degree of functionality and/or structural changes in the protein molecule. Defects in the protein molecule suggest an increased concentration of unbound drugs in the plasma, disturbances in their metabolism, and an increased risk of drug intoxication [75].

#### 3.3.4. Antioxidant Properties of Albumin

Albumin represents the largest thiol pool in the human body. Its molecule contains seventeen disulfide bonds and one free thiol group (-SH) at the cysteine residue (Cys34), which determines the involvement of albumin in modulating the redox status in the body [65,76,77,78,79]. Albumin exerts its antioxidant activity through its sulfhydryl group at Cys34-SH, which defines the key role of the protein in maintaining normal physiological redox equilibrium [80]. The sulfhydryl group presenting in HSA accounts for ~80% of all free thiols in plasma. It acts as a free radical scavenger, and the cysteine residue neutralizes multiple ROS and RNS [78]. 

#### 3.3.5. Hypoalbuminemia

Low albumin levels are due to hepatocyte content, reduced albumin synthesis, amino acid malabsorption, or increased albumin excretion and decline [81,82]. HSA concentration outside the reference range has been identified as a strong prognostic indicator of morbidity and mortality [71]. In most disease states (other than liver disease), albumin synthesis is normal or increased, and the development of hypoalbuminemia reflects an increased rate of protein turnover resulting from an increased rate of catabolism [23]. For example, the acute development of hypoalbuminemia in sepsis or trauma is due to increased capillary permeability to albumin, which leads to its redistribution from the vascular to the interstitial space [71]. Soeters et al. [73] defined low albumin levels as a function of inflammation and highlighted the involvement of various cytokines, growth factors, or hormones known to be involved in the pathogenesis of hypoalbuminemia.

#### 3.3.6. Albumin Infusion in Cases of Hypoalbuminemia

HSA is used intravenously in hypoalbuminemia to manage perioperative blood loss and restore normal cardiac output after surgery and in critically ill patients [83,84,85]. Different volume replacements are used depending on the cause of fluid loss or rational fluid therapy, including a combination of crystalloid and colloid solutions [86,87]. Synthetic colloids are characterized by relative safety and are preferred due to their low cost compared to HSA solutions. In recent years, the clinical application and therapeutic potential of HSA have shown a significant reduction in mortality and renal damage in patients with cirrhosis and peritoneal infection; therefore, the use of HSA is increasing and is preferred over crystalloid and synthetic colloidal solutions [81]. Albumin dialysis is very effective in removing non-covalently protein-bound substances. This therapeutic approach provides albumin with a high capacity to bind toxins and effectively remove them from the bloodstream. Intravenous administration of smaller infusion volumes of albumin may be sufficient for hemodynamic stabilization in patients with infections compared with crystalloid administration. This suggests that intravenous administration of 20–25% HSA solutions may improve the efficacy of renal replacement therapy in dialysis patients with hypoalbuminemia and reduce the risk of intradialytic hypotension [88,89].

### 3.4. COVID-19, Inflammation, Oxidative Stress, and Hypoalbuminemia

#### 3.4.1. Oxidative Stress and Low Albumin Level as a Function of Chronic Inflammation in COVID-19

In critical conditions, including COVID-19, inflammation is known to lead to changes in HSA levels by reducing HSA synthesis and increasing vascular permeability, allowing the protein to escape from the vascular space. This places the protein concentration outside the reference range [90]. For example, albumin production can be inhibited by the pro-inflammatory mediators IL-6, IL-1, and TNF-α [91], placing patients with COVID-19 and hypoalbuminemia at high risk of cardiovascular complications such as arterial and venous thrombotic events, arrhythmias, hypertension, atrial fibrillation and heart failure [92,93,94], longer hospital stays, and higher mortality rates [19,24,95,96].

#### 3.4.2. Overproduction of Free Radicals and Change of Reduce/Oxidize Albumin Level

Albumin actively participates in a wide range of redox reactions with pro-oxidant metals, cytotoxic reactive aldehydes, oxygen radicals, and other oxidants. It binds the free ions of transition metals (iron and copper) and thus prevents their participation in Fenton reactions with the generation of •OH. Conditions associated with ischemia, hypoxia, acidosis, and OS can transiently alter the ability of amino acid residues in albumin to bind metal ions [97]. The modification of HSA by RONS during ischemic events may be a biochemical marker of myocardial ischemia [98]. In healthy humans, about 70–80% of total albumin is in reduced form (mercaptalbumin), 25% is a non-mercaptalbumin fraction in the form of mixed disulfides between Cys34 and low molecular weight thiols [80,99], and a small fraction (~1%) oxidizes albumin to sulfinic and sulfonic acids, etc. High levels of oxidants (H_2_O_2_, ONOO^−^) or one-electron oxidation of the thiol group to a thiyl radical (R•) is accompanied by the generation of secondary radicals [76]. As a result, cysteine is oxidized to sulfenic acid (HSA-SOH), which, in the presence of ROS, can be further oxidized to sulfinic (HSA-SO_2_H) and sulfonic (HSA-SO_3_H) acids as end products of the irreversible oxidation of the protein. Under nitrosative stress, reduced cysteine is transformed into s-nitroso-albumin (HSA-SNO) [100,101]. Wybranowski et al. [24] commented on the relationship between OS and hypoalbuminemia, suggesting that redox imbalance, together with reduced albumin levels, increases the risk of death in patients with coronavirus infection. Based on the scientific information known to date, the tracking of serum albumin levels and conformational changes in the protein will allow the identification of complications in COVID-19 patients.

#### 3.4.3. Oxidative Modifications of Albumin in COVID-19

COVID-19 is characterized by a wide range of complications, which include pulmonary embolism, thromboembolism and arterial clot formation, arrhythmias, cardiomyopathy, multiorgan failure, etc. [102,103,104,105,106,107]. Disease processes such as acute respiratory distress syndrome (ARDS), lung injury, thrombosis, and cardiovascular disease (CVD) are indicative of severe COVID-19, with some patients developing pulmonary and cardiovascular complications after recovery from COVID-19 regardless of age or the presence of their pre-existing condition [108].

Oxidative stress is one of the major factors in the pathology of COVID-19 [109,110,111,112,113,114,115] and post-COVID-19 syndrome [116,117]. It is associated with extensive structural changes in multiple proteins, including albumin, suggesting the proliferation of malfunctioning derivatives of this critical protein. Induced structural and functional changes in albumin are accompanied by hypoalbuminemia [118] and can be a serious prerequisite for ineffective therapy with frequent complications and high mortality in patients with SARS-CoV-2 [119,120]. Overproduction and abnormal concentrations of free radicals, together with ineffective antioxidant defenses of the body, can be seen as leading factors in severe COVID-19 [121,122]. Under conditions of OS, proteins undergo various modifications, including disruption of their function and compromise of their biological activity. As a result, malfunctioning protein derivatives proliferate in the body [18,123]. Hyperactivation of innate immunity and overproduction of ROS are thought to be major contributors to the massive multiorgan damage caused by the SARS-CoV-2 infection [32]. High levels of pro-inflammatory cytokines, especially IL-6, initiate the release of large amounts of hepcidin during the cytokine storm, leading to the disruption of iron homeostasis through hemoglobin degradation and Fe ions dissociation [124]. It is known that hydrogen peroxide and neutrophil-mediated OS are basic factors of structural changes in HSA in critically ill COVID-19 patients [34]. At the basis of this mechanism are high concentrations of free circulating iron in the bloodstream, and together with H_2_O_2_, participates in Fenton reactions with the formation of highly reactive hydroxyl radicals [125,126,127] (Figure 3).

Cavalcanti and colleagues measured H_2_O_2_ concentration in plasma and red blood cells of patients with SARS-CoV-2 infection and found significantly reduced peroxide concentration compared to healthy volunteers. Their results suggest that H_2_O_2_ is converted into highly reactive •OH, which leads to rapid multi-organ damage, disease progression, and fatal outcomes [53]. Due to its high reduction potential and high diffusion rate, •OH participates in a wide range of redox reactions and is considered one of the most harmful and highly toxic oxidants, leading to ROS-induced oxidative damage of proteins [128]. Indeed, OC and high levels of H_2_O_2_ lead to an increased concentration of oxidized albumin, which is the less flexible and more rigid form of the protein, and a loss of biological activity of the protein [53]. These findings highlight the development of new methods for monitoring COVID-dependent hypoalbuminemia and protein dysfunction, which, according to Aziz and colleagues, is a hallmark of worsening general health and increased risk of mortality in patients with SARS-CoV-2 infection [60].

### 3.5. Redox States Analysis Based on Nitroxide Use

#### 3.5.1. Redox Sensor Properties of Cyclic Nitroxide Radical TEMPOL

For more than 50 years, stable nitroxides, also called aminoxyl radicals (>NO•), have played an important role in the theoretical analysis and experimental studies of numerous chemical and biochemical processes. Nitroxides, also known as nitroxyls, are five-membered (pyrrolidine, pyrroline, or oxazolidine compounds) or six-membered (piperidine) cyclic structures containing a nitrogen atom in their ring [129]. The presence of methyl groups in the alpha position increases the stability of nitroxides, restricts the access of reactive molecules, and prevents radical–radical dismutation reactions [130]. As redox-active species, they participate in electron transfer reactions by being reduced to their respective diamagnetic forms—hydroxylamine or oxoammonium cation [131,132]. The nitroxide radical TEMPOL is often used to analyze redox changes in biological environments. It is a redox-cyclic nitroxide that promotes the metabolism of many reactive oxygen species.

One of the first studies on TEMPOL application in humans was published in 2004 in the Clinical Cancer Research Journal. Metz et al. conducted a trial enrolling 12 patients with moderate and severe alopecia as a result of chemotherapy for lung cancer, breast cancer, melanoma, and unknown primary tumors, of which 10 completed TEMPOL treatment [133]. The alopecia treatment protocol included TEMPOL (70 mg/mL) in 70% EtOH, which was applied to the patient’s scalp 15 min before radiation (300 cGy/d). The research did not detect serious TEMPOL-related adverse events as well as abnormal biochemical and hematological laboratory blood test results, possibly due to the minimal systemic absorption of nitroxide in the blood. The study results demonstrate the prevention and protection of radiation-induced alopecia by the local administration of a TEMPOL solution [134]. The authors assumed that TEMPOL protection from radiation is due to its ability to reduce oxidation of transition metals and limit free radical damages by action as a superoxide dismutase mimetic agent [133]. TEMPOL is known to protect mitochondria from oxidative damage, improve tissue oxygenation, inhibit proliferation, and increase the vulnerability of cancer cells to apoptosis induced by cytotoxic agents [135]. It possesses pronounced SOD-mimetic activity. Like endogenous SOD, nitroxide acts as a catalyst in the dismutation process of O_2_•^−^ to H_2_O_2_ and oxygen. The reaction led to the production of oxoammonium cation that reduced in the presence of O_2_•^−^ back to a nitroxide radical. Hydroxylamines (>N-OH) can be slowly oxidized by •OH, HO_2_•, and/or Fe^3+^ ions to form the corresponding nitroxide radical [136]. In vivo and in vitro studies present the oxidized or paramagnetic form of the nitroxides as contrast, which is detected by EPR spectroscopy and nuclear magnetic resonance/magnetic resonance imaging (NMR/MRI) [137,138,139,140,141] (Figure 4).

The ability of nitroxides to participate in redox reactions in the presence of oxidants makes them suitable redox sensors for analyzing the redox status of biological systems [142,143,144]. Measuring the intensity of the EPR signal of nitroxide radicals allows for establishing changes in the redox environment of cells and tissues [138]. The balance between the two forms >NO• and >N-OH depends on the redox status of the studied medium. The paramagnetic states of the nitroxides provide EPR contrast, while the diamagnetic forms of hydroxylamine and oxoammonium cation are not detected by EPR spectroscopy [142]. TEMPOL, in combination with Ebselen, improves the bioavailability of NO and prevents the formation of the so-called uncoupling eNOS—a source of superoxide radicals—thereby protecting the endothelium from oxidation and restoring endothelial function [145]. TEMPOL was found to improve the thermal hyperemia response and effectively reverse impaired endothelial function through nitric oxide protection from generated superoxide radicals in smokers [146]. Also, it inhibits the glycation of low-density lipoprotein (LDL) and subsequently blocks glycated LDL-induced abnormal ER stress, which is associated with endothelial dysfunction [147]. Cyclic nitroxide radicals are oxidized by •OH in the presence of transition metal ions to their corresponding oxoammonium cation (ferroxidase activity of nitroxides) (Figure 5) [148].

#### 3.5.2. TEMPOL as Spin Detectors in Determination of COVID-19-Related Oxidative Stress

According to Wilcox, the reaction between TEMPOL and the O_2_•^−^ leads to the formation of an oxoammonium cation (>N+=O). This indicates that TEMPOL has two reaction sites that contribute to the efficient removal of O_2_•^−^—the nitroxide group and the fourth position of the piperidine ring [134]. Indeed, TEMPOL was determined to be an efficient catalase-like agent and is one of the best general-purpose cyclic redox sensors to prevent the generation of •OH from H_2_O_2_. Its sensitivity increases in the order O_2_•^−^ < H_2_O_2_ < •OH [149]. The rate constants of the reactions of nitroxides with •OH were found to be of the order of 10^9^ M^−1s−1^ [150]. Based on the above, it can be assumed that the change in the intensity of the nitroxide radical TEMPOL in patients with critical COVID-19 (Figure 3) is due to a recombination reaction of the nitroxide with the hydroxides generated in the presence of Fe (II) radicals and/or with secondary ROS to the formation of >N+=O as the end product of partial oxidation of the redox sensor [151]. In addition, the oxoammonium cation can be reduced in the presence of hydrogen atoms by hydrogen donors such as reduced forms of glutathione (GSH), β-nicotinamide adenine dinucleotide (NADH), or β-nicotinamide adenine dinucleotide phosphate (NADPH) to form hydroxylamine as the end product of oxidation [136]. ROS levels in patients with severe COVID-19 are significantly elevated compared to healthy controls [152]. In addition, extensively increased levels of lipid oxidation have been reported in the lung tissues of patients with COVID-19 compared to controls, suggesting high levels of oxidative stress [114]. In connection with these data, an analysis of the general redox status of patients with a severe form of COVID-19 was performed by EPR spectroscopy and can be considered a definitive and irrefutable method for proving large-scale oxidative damage in biological systems and the body as a whole.

### 3.6. Nitroxide Radicals in the Determination of COVID-19-Related Hypoalbuminemia

Hypoalbuminemia is often associated with the progression and severity of viral, bacterial, and fungal infections, with low albumin levels increasing the risk of primary and secondary infections [81]. In recent years, stable nitroxide radicals are often used to study the rotational mobility of proteins [153]. More accurate information can be obtained by using spin tags targeting a specific site in the protein that binds specifically to free cysteine residues in HSA [154]. Dramatic changes in the normal physiological balance between redHSA and its oxidized form (oxHSA) create a significant prerequisite for impaired non-enzymatic antioxidant efficiency in the body and deterioration of the condition in patients with COVID-19 [60]. Conventional techniques in the study of albumin include observation of conformational changes and protein binding capacity by EPR spectroscopy and spin labeling with spin-coated fatty acids (SLFAs). However, this approach leads to difficulties in determining which parts of the albumin molecule undergo conformational changes, as it can bind up to seven fatty acid equivalents [154]. Site-directed spin labeling (SDSL) of biomolecules has been established as a fundamental approach in molecular spectroscopy in studies of nitroxide spin-labeled proteins and DNA complexes. SDSL is a method for studying protein structure and conformational dynamics. It is based on site-directed nitroxide spin labeling by position-specific cysteine mutagenesis, especially in studies of functional protein dynamics involved in signal transduction and soluble ligand-binding proteins [155]. Introducing a paramagnetic nitroxide side chain as a molecular sensor enables the mapping of the protein skeleton by backbone fluctuations and conformational switching for relevant time intervals under an applied magnetic field [156]. Methods involving specific spin labels, such as 3-Maleimido-PROXYL, 5-DOXYL-stearic acid (from the protein interior), 16-DOXYL-stearic acid (from the protein surface) [157], etc., can be applied for rapid and accurate non-invasive diagnostics in the study of protein structures and determination of conformational changes in numerous proteins, DNA, RNA, and lipids. It is known that the measurement of the spectral parameters of the 3-Maleimido-PROXYL radical by SDSL-EPR can provide important information about the rotation of the nitroxide with respect to the applied magnetic field in the presence of proteins such as hemoglobin, albumin, etc. [154,158]. Abnormal levels of free radicals during COVID-19 may lead to oxidative modification of cysteine residues in albumin, which prevents the nitroxide radical 3-Maleimido-PROXYL from labeling the protein by forming a covalent bond with Cys-34 [7]. Low levels of albumin and structural defects in the protein suggest a risk of altered metabolism and increased plasma concentrations of unrelated drugs [159,160,161]. Based on the above, we assume that SDSL-EPR spectroscopy can be applied to various medical conditions characterized by protein dysfunction and conformational instability, including COVID-19 [7].

## 4. Limitation

SDSL is a powerful method for studying the dynamics of conformational changes in biological macromolecules, which is based on the selective grafting of paramagnetic labels at selected sites in the protein structure. Chemoselectivity is a key factor in the application of SDSL-EPR spectroscopy, as low conjugation rates or low concentrations of reaction products can lead to inefficient labeling of protein molecules. The introduction of the nitroxide spin labels during sample preparation and the stability of the formed linker, such as conformational incompatibility, can lead to inaccurate information. The reduced mobility of the label leads to a partial averaging of the anisotropy motion, while its high mobility implies a large averaging of the anisotropy motion and a narrow CW-EPR signal. Usually, EPR spectra contain multiple components, the extraction of which, especially in the case of multicomponent simulation, requires the tuning of multiple parameters. In the case of multicomponent spectra such as those observed in complex biological systems, the determination of these parameters is very difficult. Other important factors are the concentrations of the target protein and nitroxide label, the accessibility of Cys-SH groups, the different lengths of the linker and the conformational flexibility of the side chains of the spin label, the pH, and the temperature at which the assay is performed. The development of appropriate methodological protocols, the correct choice of nitroxide label, and the introduction of appropriate simulation programs could reduce these limitations and increase the applicability of EPR methods.

## 5. Conclusions

Oxidative stress and hyperinflammation are critical factors in the pathogenesis of COVID-19, and their management is critical to the treatment and prevention of the disease. Overproduction of free radicals, together with increased immune response and reduced endogenous enzymatic and non-enzymatic antioxidant defenses, lead to a vicious cycle of mutually induced hyperinflammation and massive oxidative damage. The reported abnormal levels of ROS and OS suggest a high degree of structural changes in the protein molecule, the formation of non-functional protein derivatives, and a high concentration of oxidized albumin in critical COVID-19. The infection is characterized by a long and systemic OS, with persistently high levels of ROS, oxidized albumin, and hypoalbuminemia, making it impossible to limit COVID-19-induced oxidative damage. These pathologies put the body in a state of “oxidative shock”, which is considered one of the probable causes of large-scale damage to the body. Based on this, the simultaneous monitoring of the levels of oxidized albumin and oxidants, in particular, can be of particular clinical and diagnostic importance, and oxidative stress as a marker of severe coronavirus infection. Their antioxidant properties and reaction with free radicals, such as superoxide radicals, H_2_O_2_, and other ROS, mark their potential use in biomedicine to reduce OS and its complications. As contrast agents in EPR spectroscopy, nitroxides are the best tools for scanning and regions of interest visualization in globular proteins, DNA, lipid membrane proteins, and total and local oxidative stress and antioxidant capacity monitoring.

## 6. Patents

The patents resulting from the work reported in this manuscript are identified by the incoming number of the patent application: BG/U/2022/5487.

## Figures and Tables

**Figure 1 ijms-25-08045-f001:**
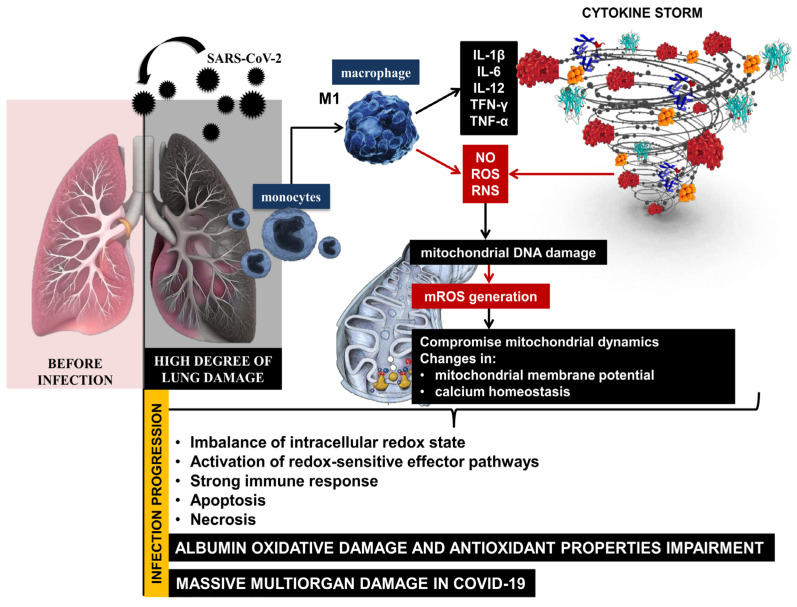
Immune response and redox imbalance during SARS-CoV-2 infection.

**Figure 2 ijms-25-08045-f002:**
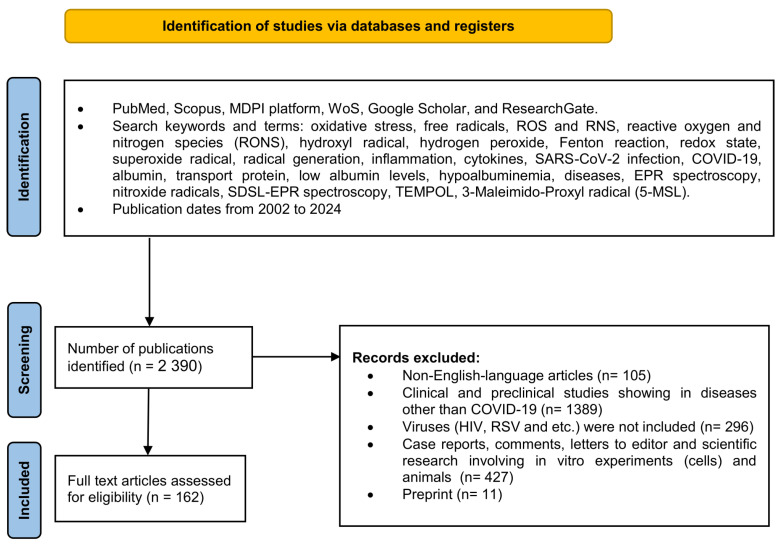
The flow diagram shows the methodology for selecting articles. This methodology follows the recommendations given in PRISMA-P guideline rules [25] and PRISMA-S named “PRISMA-S: an extension to the PRISMA Statement for Reporting Literature Searches in Systematic Reviews” for reporting literature searches.

**Figure 3 ijms-25-08045-f003:**
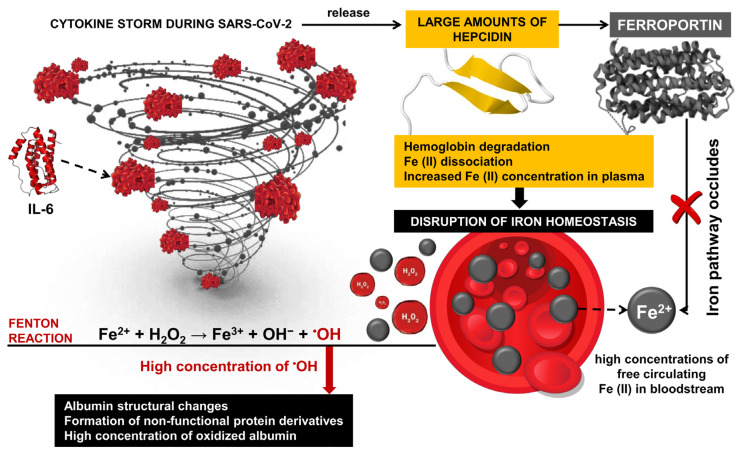
Inflammation response and hepcidin during SARS-CoV-2 infection. Transported iron into cells (Fe^2+^) binds to ferritin or is transported into the bloodstream by ferroportin, where it is oxidized to Fe^3+^. Oxidized iron forms a complex with transferrin, which allows efficient iron transport in the body. The inflammatory response triggers the production of hepcidin, which is part of the body’s defense mechanism against pathogens. During SARS-CoV-2 infection, interleukin-6 (IL-6) induced hepcidin induction via the IL-6R/STAT3 pathway. It leads to high hepcidin levels and, as a result, to inhibiting and decreasing ferroportin activity.

**Figure 4 ijms-25-08045-f004:**
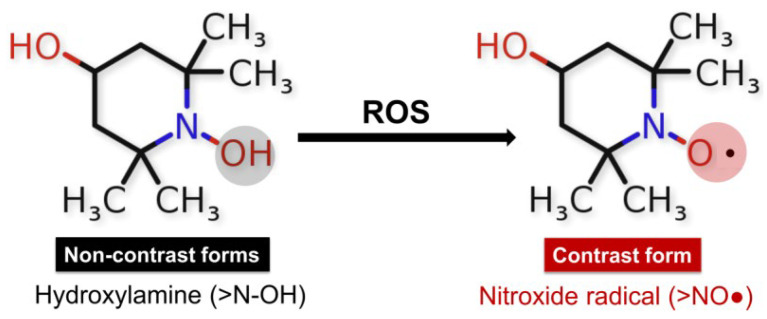
Redox reactions of nitroxides: diamagnetic non-radical forms hydroxyl amine (>N-OH), and paramagnetic radical form (>NO•) involvement of the hydroxylamine form of TEMPOL-H in redox reactions in the presence of ROS, in which the emergence of an EPR signal and recovery of the nitroxide radical form is observed.

**Figure 5 ijms-25-08045-f005:**
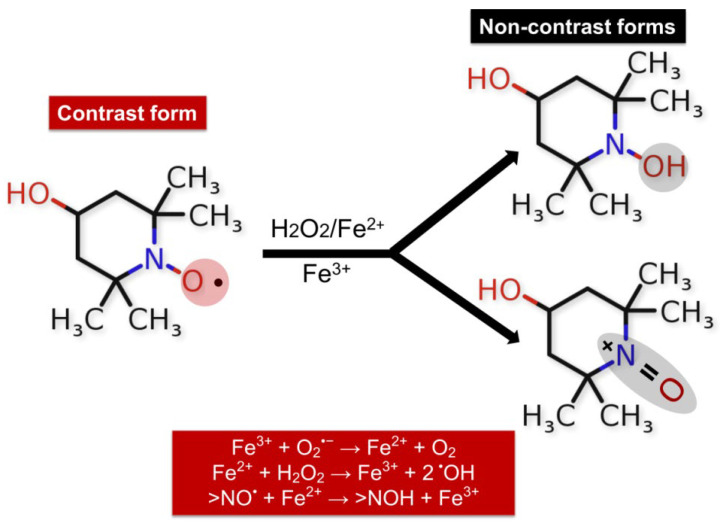
Redox reactions of TEMPOL in the presence of free iron, •OH, and H_2_O_2_. Efficiency of the nitroxide radical in metabolizing O_2_•^−^ and H_2_O_2_ or in protecting cells from •OH; TEMPOL sensitivity was found to be the highest for •OH.

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
