# Peer review of "Stable Nitroxide as Diagnostic Tools for Monitoring of Oxidative Stress and Hypoalbuminemia in the Context of COVID-19"

_ijms, 2024, doi:10.3390/ijms25158045_

Round 1

Reviewer 1 Report

Comments and Suggestions for Authors

This is an interesting review presenting the interrelationship between oxidative stress, inflammation, and low albumin levels, particularly in patients with SARS-CoV-2 infection.

Comments:

1.     It is well demonstrated that Serum Albumin Levels is a biomarker in different disease settings, as indicated in bibliographic citations 12 (line 67) and 13 (line 72). But it cannot be stated that the decrease in albumin concentration is the main indicator of the severity of the COVID-19 infection, because there are other indicators of the same importance. Therefore, the phrase that appears in the Introduction section should be formulated in a less categorical way: "Therefore, albumin values ​​below 3.5 g/dL or 35 g/L 70 should be considered the main indicator determining the severity of SARS-COV- 2 infection and disease outcome".(lines 71-72). 2.     It is not yet widely accepted that Albumin Oxidation plays a relevant role in different diseases. In fact, this finding has been described in patients with COVID-19 Pneumonia (Wybranowski, Int. J. Mol. Sci. 2023), and is not extensible to any patient with SARS- COV-2 infection. Therefore, it cannot be stated that "oxidized albumin is a marker of SARS-CoV-2 infection" (lines 81-82) This marker is not sufficiently studied in the large population of patients with SARS-COV-2 infection, which is very complex and varied. 3.     At the end of the Discussion the following phrase: "Structural-functional changes in albumin accompanied by hypoalbuminemia can be a serious prerequisite for ineffective therapy, frequent complications, and high mortality in SARS-CoV-2 infection" (lines 427-429) does not seem to fit the clinical data presented in different publications. It cannot be said that hypoalbuminemia and an imbalance between reduced and oxidized albumin are really a serious prerequisite for ineffective therapy, and high mortality in SARS-CoV-2 infection

4.     The term “COVID-19 infection” is used throughout the manuscript, and the term SARS-CoV-2 infection is used a few times. Perhaps it would be more appropriate to use the same terminology in all cases, and currently the expression SARS-CoV-2 is used more.

Comments on the Quality of English Language

Minor editing English is required

Author Response

RESPONSES TO THE Reviewers' COMMENTS

We appreciate reviewers’ comments. All corrections in the manuscript are highlighted in green.

Comment 1.     It is well demonstrated that Serum Albumin Levels is a biomarker in different disease settings, as indicated in bibliographic citations 12 (line 67) and 13 (line 72). But it cannot be stated that the decrease in albumin concentration is the main indicator of the severity of the COVID-19 infection, because there are other indicators of the same importance. Therefore, the phrase that appears in the Introduction section should be formulated in a less categorical way: "Therefore, albumin values ​​below 3.5 g/dL or 35 g/L 70 should be considered the main indicator determining the severity of SARS-COV- 2 infection and disease outcome".(lines 71-72).

Response 1: We paraphrased the sentence as follow: Therefore, albumin values ​​below 3.5 g/dL or 35 g/L might be considered as an indicator determining the severity of SARS-CoV-2 infection and disease outcome

Comment 2.      It is not yet widely accepted that Albumin Oxidation plays a relevant role in different diseases. In fact, this finding has been described in patients with COVID-19 Pneumonia (Wybranowski, Int. J. Mol. Sci. 2023), and is not extensible to any patient with SARS- COV-2 infection. Therefore, it cannot be stated that "oxidized albumin is a marker of SARS-CoV-2 infection" (lines 81-82). This marker is not sufficiently studied in the large population of patients with SARS-COV-2 infection, which is very complex and varied.

Response 2: We agree with the reviewer and deleted the sentence from line 81-82.

Comment 3.     At the end of the Discussion the following phrase: "Structural-functional changes in albumin accompanied by hypoalbuminemia can be a serious prerequisite for ineffective therapy, frequent complications, and high mortality in SARS-CoV-2 infection" (lines 427-429) does not seem to fit the clinical data presented in different publications. It cannot be said that hypoalbuminemia and an imbalance between reduced and oxidized albumin are really a serious prerequisite for ineffective therapy, and high mortality in SARS-CoV-2 infection

Response 3: This conclusion is in response to our research on the concentrations of the therapeutic drugs with which the patients with covid 19 were treated. The results show that all the drugs are in concentrations lower than the therapeutic ones. These results are about to be published, in which case we will modify the sentence according to your recommendations.

Comments 4: The term “COVID-19 infection” is used throughout the manuscript, and the term SARS-CoV-2 infection is used a few times. Perhaps it would be more appropriate to use the same terminology in all cases, and currently the expression SARS-CoV-2 is used more.

Response 4:      Depending on the context of the sentence, we have used both terminologies, with COVID-19 we mean the disease, and with the infection we mean the infection caused by „SARS-CoV-2 infection”

Reviewer 2 Report

Comments and Suggestions for Authors

Congratulation for your laborious efforts to prepare the manuscript in order significant information to the field to be offered to the scientific community. However, I have serious doubts of the way your manuscript will provide a clear view of the severe covid-19 pneumonia pathophysiologic conditions and mechanisms processed at a molecular level.

I would then like to comment about the following issues

Comment 1: More specifically you state as a title “Stable Nitroxide as a diagnostic tool(s) for monitoring of oxidative stress and hypoalbuminemia in the context of covid-19”, but nowhere in the text is described any certain way of monitoring of the referred oxidative species and relevant significantly statistical evaluation for correlation with hypoalbuminemia in severe covid-19 disease.

Comment 2: Your review contains a significant amount of current knowledge about the biochemical disturbances in sepsis but not clear association of the redox state with hypoalbuminemia. Moreover, severe disease leads initially to an upregulation of the hepatic protein production (acute phase proteins) which is indicative of the disordered metabolism before a level of hepatic failure will be established. Thus, we are not aware about the remarks and the description of the severe Covid-19 condition collected from the papers you had reviewed,

Comment 3: There are not included in your review studies with amino-acid stable isotopes where the oxidation and transamination rates will give us information of the kinetics of proteins accurately.

Comment 4: The manuscript, when revised, must contains with separate paragraphs, all the relevant information about the redox processes in normal life,  in diseases, in sepsis and extensively  in Covid-19 disease with elimination of any possible assumptions.

Comment 5: You must distinguish the roles of the oxidative specie as mitochondrial, intracellular extracellular oxidative biomoleciules and environmental when you refer to oxidative stress relevant to Covid-19 disease.

Comment 6: We have no information about the role and the depletion of any endogenous antioxidants

Comment 7: When an oxidative specie like nitrogen is proposed to serve as a surrogate marker, we need kinetics and significant statistical correlations to proposed biochemical indicator with clinical significance.

Comment 8: You have to enrich and focus the content of your manuscript to information about the Covid-19 pathophysiology traits and to discuss any possible limitations regarding the conclusion of the papers you included in your review

Comments on the Quality of English Language

Extensive editing is needed by a natively speaking the Eglish language

Author Response

RESPONSES TO THE Reviewers' COMMENTS

We appreciate reviewers’ comments. All corrections in the manuscript is in red.

Comment 1: More specifically you state as a title “Stable Nitroxide as a diagnostic tool(s) for monitoring of oxidative stress and hypoalbuminemia in the context of covid-19”, but nowhere in the text is described any certain way of monitoring of the referred oxidative species and relevant significantly statistical evaluation for correlation with hypoalbuminemia in severe covid-19 disease.

Response 1:

            Previous studies of our collective "Site-Directed Spin Labeling EPR Spectroscopy for Determination of Albumin Structural Damage and Hypoalbuminemia in Critical COVID-19" and "Direct Application of 3-Maleimido-PROXYL for Proving Hypoalbuminemia in Cases of SARS-CoV-2 Infection: The Potential Diagnostic Method of Determining Albumin Instability and Oxidized Protein Level in Severe COVID-19” present a method for monitoring total oxidative stress, hypoalbuminemia and oxidized/reduced albumin levels in patients with moderate and severe COVID-19 [Georgieva et al, 2022; Georgieva et al., 2023]. According to the obtained results, it can be assumed that the nitroxide radicals TEMPOL and 3-Meleimido-PROXYL are reliable reporters of general oxidative stress and conformational changes in the albumin molecule. The obtained biophysical parameters for nitroxide radicals show a positive correlation with the severity and mortality in the studied patients and justify the application of EPR spectroscopy in patients with this disease. Our preliminary patient studies show that nitroxide-enhanced EPR spectroscopy may be a promising method for staging patients with COVID-19, assessing structural-dynamic changes in protein systems, and protein instability, and determining the ox/redHSA ratio that are out of the scope of conventional techniques.

Comment 2: Your review contains a significant amount of current knowledge about the biochemical disturbances in sepsis but not clear association of the redox state with hypoalbuminemia. Moreover, severe disease leads initially to an up regulation of the hepatic protein production (acute phase proteins) which is indicative of the disordered metabolism before a level of hepatic failure will be established. Thus, we are not aware about the remarks and the description of the severe Covid-19 condition collected from the papers you had reviewed.

Response 2:

            A systemic inflammatory response known as cytokine release syndrome or cytokine storm plays a major role in developing acute injury during COVID-19. A rapid increase in the levels of pro-inflammatory cytokines in SARS-CoV-2 infection is associated with the development of acute respiratory distress syndrome (ARDS) [Montazersaheb et al., 2022]. In severe COVID-19, ARDS is defined as a predictable serious complication that requires early recognition. The entry of the SARS-CoV-2 virus into the respiratory system and the subsequent strong inflammatory response leads to the destruction of the alveolar-capillary barrier [Georgieva et al., 2023]. In the acute stage of infection, ARDS causes lung damage, which includes the formation of hyaline membranes in the alveoli, followed by interstitial expansion, fibroblast proliferation, typical pathological changes characterized by diffuse alveolar parenchymal damage and edema [Georgieva et al., 2023]. Diffuse alveolar damage results in impaired gas exchange, with refractory hypoxemia and hypercarbia, intrapulmonary shunt, and reduced functional lung surface [Dushianthan et al., 2023]. Accumulation of fluid in the alveolar and interstitial spaces causes inhibition of pulmonary surfactant [Tlatelpa-Romero, B et al., 2023]. The lung morphology is characterized by a rapid evolution from interstitial and alveolar edema to fibrosis, as a result of damage to the alveolar-capillary unit [Savin et al., 2022]. Viral infections are known to be characterized by the production of abnormally high levels of oxygen radicals. For example, SARS-CoV-2 causes overactivation of the immune response in lung tissues, which is almost always accompanied by oxidative stress and subsequent endothelial damage [Georgieva et al., 2023]. ROS production can cause oxidative damage to lung tissue, which together with the inflammatory response, leads to massive lung dysfunction and impaired oxygen exchange. Redox imbalance further contributes to the progression of ARDS and the development of respiratory failure [Bezerra et al., 2023 [6]]. Excess ROS causes irreversible oxidative damage to important biomacromolecules, membrane phospholipids, and proteins [Pantelis et al., 2023]. In systemic inflammatory processes, there are deviations from the reference values ​​of various biochemical indicators, which is complemented by another important prognostic marker - serum albumin. HAS is affected by various factors, such as changes in the extracellular redox balance in favor of oxidants, which can lead to structural disorders in its molecule, pH, transport of transition metal ions, nitric oxide, hemin, and drugs. In the acute phase of COVID-19, HAS acts as an antioxidant, but high levels of OS can lead to its irreversible oxidation [Georgieva et al., 2022]. The overproduction of free radicals, together with an increased immune response and reduced endogenous enzymatic and non-enzymatic antioxidant defenses, lead to a vicious cycle of mutually induced hyperinflammation and massive oxidative damage during the acute phase of COVID-19. This implies a high degree of structural changes in macromolecules, the formation of non-functional protein derivatives, and a high degree of damage to cellular components [Georgieva et al, 2023].

Comment 3: There are not included in your review studies with amino-acid stable isotopes where the oxidation and transamination rates will give us information of the kinetics of proteins accurately.

Response 3:

            The most commonly used biophysical techniques to study the structural and dynamic properties of membrane proteins are X-ray crystallography, nuclear magnetic resonance (NMR), electron microscopy, and Förster resonance energy transfer. However, they have several limitations, including the type of medium, the size of the proteins, the size of the probe, etc. In recent years, EPR spectroscopy has been increasingly used as a biophysical tool to study the structure and dynamics of various membrane proteins, as it minimizes these limitations. SDSL-EPR spectroscopy can be used to monitor transitions from induced structural changes and possible conformational modifications due to protein-nucleotide and protein-protein interactions. Despite the wide application of EPR spectroscopy, MS, HPLC-MS/MS, and NMR are more suitable for following the kinetics of protein oxidation and transamination. Amino acid stable isotopes are often used in mass spectrometry and NMR to track the metabolism and dynamics of proteins within biological systems. These techniques involve labeling specific atoms in biomolecules, such as carbon, nitrogen, or hydrogen, with stable isotopes such as 13C, 15N, or 2H. They are particularly applicable in the study of protein turnover rates, protein-protein interactions, and metabolic pathways. The methods allow studying the rates of oxidation and transamination, the kinetics of various proteins, and the overall dynamics of protein synthesis and degradation can be followed. MS, HPLC-MS/MS and NMR methods are not the subject of this review but could be described in future studies.

            In conclusion, specific stable isotope markers enable the monitoring of protein, lipid, and carbohydrate metabolism in acute or chronic diseases, and are subject to MS; HPLC-MS/MS, and NMR analysis, while the stable nitroxide radicals used in SDSL-EPR allow the study of the structure and dynamics of different proteins, mapping of protein molecules, measurement of inter-molecular distances and protein-protein interactions.

Comment 4: The manuscript, when revised, must contains with separate paragraphs, all the relevant information about the redox processes in normal life,  in diseases, in sepsis and extensively  in Covid-19 disease with elimination of any possible assumptions.

Response 4:

            We agree with the reviewer to include a paragraph containing information about redox processes in normal life, but there is enough information about the involvement of oxidative stress in different diseases their complication, and sepsis. The information in this overview is the COVID-19 disease. Any additional information would take the focus off the topic.

            Reactive oxygen species (ROS) are produced as a natural response from normal oxygen metabolism in the body, are involved in many cellular signaling pathways, and are required to eliminate viruses phagocytosed by immune cells. At physiological levels, ROS function as redox mediators, participating in signal transduction, and promoting cell proliferation and cell survival, while high levels of ROS can induce cell death [Gusev, and Zhuravleva, 2022]. Depending on the concentration of ROS, it can be beneficial or harmful to cells and tissues. A healthy organism is in a state of equilibrium, characterized by the maintenance of physiological levels of ROS by intracellular reductants. Under physiological conditions, the balance between ROS production and scavenging is the main factor responsible for maintaining redox homeostasis, ensuring that cells will respond properly to endogenous and exogenous stimuli. With the so-called "steady state", intracellular ROS levels are tightly regulated by antioxidant enzymes that maintain cellular redox homeostasis [Korczowska-Łącka et al., 2023]. However, disturbances in redox homeostasis and induction of oxidative stress lead to some pathological processes, abnormal cell death, and the development of various diseases. Oxidative stress serves not only as a type of stimulus to induce a transduction response but can also modulate apoptosis through direct modifications of biological macromolecules [Vona et al., 2023]. When oxidative stress occurs, cells try to counteract it by restoring the redox balance and regulating critical homeostatic parameters. Such cellular activity results in the activation or inactivation of genes encoding defense enzymes, transcription factors, and structural proteins [Vona et al., 2023]. ROS are involved in cell signaling in the regulation of cellular processes, thanks to their ability to mediate the reversible oxidation of cysteine [Gusev, and Zhuravleva, 2022]. For example, H2O2 has emerged as the major redox-signaling metabolite capable of mediating the reversible oxidation of thiol groups in proteins.

Comment 5: You must distinguish the roles of the oxidative specie as mitochondrial, intracellular extracellular oxidative biomolecules and environmental when you refer to oxidative stress relevant to Covid-19 disease.

Response 5:

            The superoxide anion radical (O2•‾) is generated in cells by enzymatic and non-enzymatic processes. Enzymatic sources include NADPH oxidase activity, xanthine oxidase, the mitochondrial electron transport chain, and some enzymes in peroxisomes [Andrés et al, 2023]. Non-enzymatic sources can be redox reactions involving metals such as iron and copper and external sources such as toxins and xenobiotics radiation, and some drugs. The superoxide anion radical can react with another superoxide to generate hydrogen peroxide (H2O2), which can be reduced to water or partially reduced to the highly reactive hydroxide radical (•OH) [Martemucci et al., 2022]. In turn, superoxide anion dismutation can be spontaneous or catalyzed by enzymes known as superoxide dismutases (SODs). The formation of •OH is possible by the decomposition of H2O2, in the presence of ions of transition metals in a lower valence state (Fe2+ or Cu+). An important mechanism for the generation of hydroxide radicals is the reaction of hydrogen peroxide and superoxide radical (Haber-Weiss reaction) in the presence of oxidized transition metals [Martemucci et al, 2022]. In biological systems, O2•‾ is also a major precursor of other highly reactive species. Its reaction with nitric oxide (NO) leads to the formation of the highly reactive peroxynitrite (ONOO-), which is actively involved in OC and inflammation. The interaction of O2•‾ and NO can lead to reduced nitric oxide bioavailability, affecting endothelial function [Martemucci et al., 2022].

            Mitochondrial dysfunction has been proposed as a potential mechanism in the pathology of COVID-19. Mitochondrial redox control is important not only for oxidative phosphorylation, ATP synthesis, calcium homeostasis, thermogenesis, apoptosis, and ROS production but also for maintaining redox balance in cells [E Georgieva, et al., 2023]. Mitochondrial gene mutations, which underlie various diseases, can disrupt mitochondrial energy metabolism, mitochondrial bioenergetics, and biosynthesis and serve as a trigger for mitochondrial “retrograde signaling” in the nucleus. disorders in redox control define mitochondria as the main source of intracellular oxidants [E Georgieva, et al., 2023]. Pathological changes in mitochondrial dynamics can be caused by the overproduction of ROS and the mitochondrial dysfunction initiated by them. As a result, processes of oxidative DNA damage are promoted and a prerequisite is created for impaired redox regulation, which contributes to a wide range of pathological changes in cells [Juan et al., 2021]. Several studies have reported that SARS-CoV-2 can lead to mitochondrial dysfunction in various cell types. The virus induces a significant decrease in mitochondrial membrane potential and increased ROS production in lung epithelial cells, which may lead to OS and lung tissue damage [Andrieux et al., 2021]. Direct evidence for this is found to be higher levels of mtDNA in the blood of patients with COVID-19, which is due to increased mitochondrial stress (mtOS) and mitochondrial dysfunction [E Georgieva, et al., 2023]. The redox couple GSH/GSSG is fundamental for cells and plays a key role in the regulation of redox-dependent cellular functions – through thiol modifications. As a key modulator of cellular functions, GSH is involved in cellular defense against oxidative damage, in the redox regulation of protein thiols and maintenance of cellular redox homeostasis in nutrient metabolism, and the regulation of cellular metabolic functions ranging from gene expression, protein synthesis, signal transduction to cell proliferation and apoptosis [Musaogullari, et., 2020]. Any change in the redox balance of the cell in favor of the oxidized form of glutathione–GSSG, represents an important signal that can determine the fate of the cell [Marí et al., 2020]. Depletion of mitochondrial GSH can lead to increased release of H2O2 from the matrix, which can oxidize cytoplasmic proteins and affect cell signaling [Kowalczyk et al., 2021]. In conditions of prolonged oxidative stress, when cellular defense systems cannot counteract oxidative-mediated damage, the amount of free GSH decreases. This leads to irreversible cell degeneration and death [Vašková et al., 2023].

            Reduced plasma GSH levels have been observed in patients with COVID-19. The increased oxidation of cellular components such as the ROS/GSH ratio in favor of oxidants strongly correlates with the severity, symptoms, and recovery period of infection [Kumar, P et al., 2022]. Protein glutathionylation is the main redox immune mechanism that prevents or alleviates damage to cellular components [Žarković et al, 2022]. In patients with COVID-19, systemic oxidative stress leads to a decrease in GSH levels, promoting the development of infection, while an increase in GSH levels inhibits disease progression. Despite the increased activity of enzymes, especially glutathione peroxidase, low levels of GSH inhibit the effectiveness of non-enzymatic antioxidant activity, especially in those who died as a result of COVID-19 [Žarković, et al., 2022; Kumar, P et al., 2022; Silvagno, F et al., 2020]. NADPH is an important cofactor involved in many physiological processes and disturbances in its synthesis, as well as an imbalance in the ratio of reduced/oxidized forms leads to the development of pathologies. In the mitochondria, the redox couple NADPH/NADP+ under the action of nicotinamide nucleotide transhydrogenase in combination with NADH/NAD+ leads to the accumulation of NADPH. Any disturbance in the components of this enzyme could lead to changes in the redox state and the appearance of oxidative damage in mitochondrial structures. NADPH-dependent oxidases use NADPH as a cofactor and are thought to be the only group of enzymes for which ROS production is a major function. They are membrane-bound enzymes that generate superoxide and other ROS (H2O2) at the plasma membrane. Disturbances in NADPH production are expected to modulate cellular redox balance and lead to increased oxidation [Curtis et al., 2022]. Increased oxidative stress through Nox2 activation is associated with severe disease and thrombotic events in COVID-19. ROS production and S protein-induced activation of NOX2 in endothelial cells initiates endothelial dysfunction in cardiac microvascular endothelium in deceased patients with COVID-19 [Morawietz, H et al, 2023]. SARS-CoV-2 generates downregulation of angiotensin-converting enzyme 2 (ACE2) and transmembrane protease serine 2 (TMPRSS2) receptors, reducing their number, leading to increased activation of angiotensin II and decreased levels of angiotensin. Binding of the virus to the ACE2 receptor causes excessive release of various inflammatory cytokines, followed by disturbances in the regulation of the renin-angiotensin-aldosterone system, activation of NADPH oxidase, progression of infection and coagulation disorders [Duloquin, G et al., 2024].

Comment 6: We have no information about the role and the depletion of any endogenous antioxidants

Response 6:

The body's defense against oxidative damage involves strict redox control and maintenance of a delicate balance between the formation and elimination of ROS and RNS [Aranda-Rivera et al., 2022]. To counteract the harmful effects of free radicals, the body has an endogenous antioxidant defense system including SOD, CAT, GPx, and glutathione reductase [Vona et al., 2023]. SOD catalyzes the conversion of the highly reactive superoxide radical (O2•–) to the less reactive hydrogen peroxide through redox reactions involving metal ions of variable valence [Napolitano et al., 2022]. Catalase is a hemoprotein involved in the metabolism of hydrogen peroxide, degrading it to water and molecular oxygen. Catalase shares this function with glutathione peroxidases. Other enzymes involved in ROS scavenging are glutathione-S-transferases (GSTs) шSzechyńska-Hebda et al., 2022]. They are involved in the metabolism of xenobiotics and the synthesis of some endogenous biologically important compounds. Antioxidants such as GSH and the endogenous antioxidants SOD, CAT, and GPX play a leading role in preventing oxidative damage in COVID-19. In patients with SARS-CoV-2 and high levels of ROS, a higher activity of CAT and SOD enzymes was observed, which correlated with the severity of the disease. At the same time, low levels of reduced thiol and reduced total antioxidant capacity are observed, especially in hospitalized patients and those with a severe form of infection [Anwar et al., 2024].

Comment 7: When oxidative species like nitrogen is proposed to serve as a surrogate marker, we need kinetics and significant statistical correlations to proposed biochemical indicator with clinical significance.

Response 7:

Nitric oxide (NO) plays a major role as a signaling molecule in the regulation of vascular tone and blood flow. It is also involved in tissue oxygenation by controlling mitochondrial O2 consumption by inhibiting cytochrome c oxidase [Habib et al., 2011]. In addition to endothelial cells, NO can be generated by immune cells in response to inflammatory stimuli, leading to protection against infections. This also determines its function as a regulatory molecule in the modulation of inflammatory reactions and the activation of the body's immune response to pathogens [Kotlyarov, 2022]. The reactivity of nitric oxide is quite limited and therefore its direct toxicity to the organism is less than that of most ROS. However, nitric oxide reacts with O2•‾ to the highly aggressive and toxic proteins, lipids, and DNA, the peroxynitrite anion (ONOO‾), which causes disruption of redox signaling and compromises the maintenance of cellular redox balance. Redox-active species derived from NO react rapidly with fatty acids and lipids, generating oxidized and nitrated products of free lipids and esterified cholesterol [Korczowska-Łącka et al.,2023].

Peroxynitrite (ONOO‾) is a highly reactive powerful oxidizing and nitrating agent. In biological systems, it is formed by a rapid diffusion-controlled reaction between nitric oxide and O2•‾ [Fujii, Osaki,  2023]. The presence of transition metals such as iron or copper can speed up the process. It is characterized by strong reactivity, which through oxidation, nitration, and modification of various biomolecules, causes nitrosative stress in various types of cells. Nitration of proteins can initiate structural modifications and alter the activity of numerous biomolecules, contributing to cellular dysfunction and disruption of normal physiological processes in the body [Demirci-Cekic et al., 2022].

In severe COVID-19, IL-1 and IL-2 are the main interleukins involved in RNS production. The main reactive nitrogen species (RNS) that have been identified as mediators of redox reactions in patients with COVID-19 are nitric oxide (NO•), nitrogen dioxide radical (ONO•), and peroxynitrite radical (ONOO•). During the cytokine storm, the induced enzyme nitric oxide synthetase (NOs) stimulates the generation of nitric oxide. the NO formed reacts with the superoxide anion, resulting in the formation of the highly reactive peroxynitrite radical [Ahmed, S.A et al, 2023]. Peroxynitrite has a major role in tissue damage and inflammation in various diseases. It participates in inflammation by modulating the function of immune cells and the release of pro-inflammatory cytokines and chemokines [Pérez de la Lastra et al., 2022]. The mechanism involves the parallel formation of large amounts of "inflammatory NO" and (O2•‾, which together form ONOO‾, which initiates DNA damage, lipid oxidation, tyrosine nitration, inhibition of mitochondrial respiration [Pérez de la Lastra et al., 2022; Mandal, S.M 2023].

Comment 8: You have to enrich and focus the content of your manuscript to information about the Covid-19 pathophysiology traits and to discuss any possible limitations regarding the conclusion of the papers you included in your review.

Response 8:

SDSL is a powerful method for studying the dynamics of conformational changes in biological macromolecules, which is based on the selective grafting of paramagnetic labels at selected sites in the protein structure. Chemoselectivity is a key factor in the application of SDSL-EPR spectroscopy, as low conjugation rates or low concentrations of reaction products can lead to inefficient labeling of protein molecules. The introduction of the nitroxide spin labels during sample preparation and the stability of the formed linker, such as conformational incompatibility can lead to inaccurate information. The reduced mobility of the label leads to a partial averaging of the anisotropy motion, while its high mobility implies a large averaging of the anisotropy motion and a narrow CW-EPR signal. Usually, EPR spectra contain multiple components, the extraction of which, especially in the case of multicomponent simulation, requires the tuning of multiple parameters. In the case of multicomponent spectra such as those observed in complex biological systems, the determination of these parameters is very difficult. Other important factors are the concentrations of the target protein and nitroxide label, the accessibility of Cys-SH groups, the different lengths of the linker and the conformational flexibility of the side chains of the spin label, the pH, and the temperature at which the assay is performed. The development of appropriate methodological protocols, the correct choice of nitroxide label, and the introduction of appropriate simulation programs could reduce these limitations and increase the applicability of EPR methods.

Reviewer 3 Report

Comments and Suggestions for Authors

I have received for review a commentary entitled “STABLE NITROXIDE AS DIAGNOSTIC TOOLS FOR MONITORING OF OXIDATIVE STRESS AND HYPOALBUMINEMIA IN THE CONTEXT OF COVID-19” which is being processed by the journal IJMS.

I would like to congratulate the collective of authors for the proposed manuscript. It is an extremely interesting one, with therapeutic and prognostic value for patients with COVID-19 and long-COVID. Authors should pay attention to the following aspects in order to improve the proposed manuscript:

Rows 67-72 - in addition to the cited article, there is other evidence in the literature to support the hypothesis under consideration

Row 90 - figure 2 is missing

The discussion section provides a comprehensive review of the current level of knowledge in the literature

I suggest the authors insert an additional paragraph on potential clinical implications, scientific evidence in this regard.

I congratulate the authors for the iconography that facilitates the reading of the manuscript.

Rows 487 - 503 - if not applicable, can be omitted.

In conclusion, the proposed manuscript brings to attention an extremely interesting topic, presenting scientific information with future therapeutic value. 

Author Response

RESPONSES TO THE Reviewers' COMMENTS

We appreciate reviewers’ comments. All corrections in the manuscript are highlighted in yellow.

I have received for review a commentary entitled “STABLE NITROXIDE AS DIAGNOSTIC TOOLS FOR MONITORING OF OXIDATIVE STRESS AND HYPOALBUMINEMIA IN THE CONTEXT OF COVID-19” which is being processed by the journal IJMS.

I would like to congratulate the collective of authors for the proposed manuscript. It is an extremely interesting one, with therapeutic and prognostic value for patients with COVID-19 and long-COVID. Authors should pay attention to the following aspects in order to improve the proposed manuscript:

Comment 1: Rows 67-72 - in addition to the cited article, there is other evidence in the literature to support the hypothesis under consideration

Response 1: done

Comment 2: Row 90 - figure 2 is missing

Response: done

The discussion section provides a comprehensive review of the current level of knowledge in the literature

Comment 3: I suggest the authors insert an additional paragraph on potential clinical implications, scientific evidence in this regard.

Response 3: done

I congratulate the authors for the iconography that facilitates the reading of the manuscript.

Comment 4: Rows 487 - 503 - if not applicable, can be omitted.

Response 4: done

 In conclusion, the proposed manuscript brings to attention an extremely interesting topic, presenting scientific information with future therapeutic value.

Round 2

Reviewer 2 Report

Comments and Suggestions for Authors

The content of the manuscript is now more clear and can be considered as an effort your work to be reviewed and to summarize conclusively your findings

Comments on the Quality of English Language

The use of English language is now acceptable